# Iron Dysregulation in Alzheimer’s Disease: LA-ICP-MS Bioimaging of the Distribution of Iron and Ferroportin in the CA1 Region of the Human Hippocampus

**DOI:** 10.3390/biom14030295

**Published:** 2024-03-01

**Authors:** Susana Junceda, María Cruz-Alonso, Beatriz Fernandez, Rosario Pereiro, Eva Martínez-Pinilla, Ana Navarro

**Affiliations:** 1Servicio de Anatomo-Patología, Hospital Universitario Central de Asturias–HUCA, Av Roma s/n, 33011 Oviedo, Spain; susanajunceda@gmail.com; 2Instituto de Neurociencias del Principado de Asturias (INEUROPA), 33006 Oviedo, Spain; 3Instituto de Investigación Sanitaria del Principado de Asturias (ISPA), 33011 Oviedo, Spain; 4Department of Physical and Analytical Chemistry, Faculty of Chemistry, University of Oviedo, Julian Claveria 8, 33006 Oviedo, Spain; cruz.alonso.maria@gmail.com (M.C.-A.); fernandezbeatriz@uniovi.es (B.F.); mrpereiro@uniovi.es (R.P.); 5Department of Morphology and Cell Biology, Faculty of Medicine, University of Oviedo, Julian Claveria 6, 33006 Oviedo, Spain; martinezeva@uniovi.es (E.M.-P.); anavarro@uniovi.es (A.N.)

**Keywords:** Alzheimer’s disease, iron transport proteins, ferroportin, iron homeostasis, β-amyloid, Tau, immunohistochemistry, LA-ICP-MS

## Abstract

Alzheimer’s disease (AD) is a prevalent neurodegenerative disorder characterized by cognitive decline and neuropathological hallmarks, including β-amyloid (Aβ) plaques, Tau tangles, synaptic dysfunction and neurodegeneration. Emerging evidence suggests that abnormal iron (Fe) metabolism plays a role in AD pathogenesis, but the precise spatial distribution of the Fe and its transporters, such as ferroportin (FPN), within affected brain regions remains poorly understood. This study investigates the distribution of Fe and FPN in the CA1 region of the human hippocampus in AD patients with a micrometer lateral resolution using laser ablation inductively coupled plasma mass spectrometry (LA-ICP-MS). For this purpose, we visualized and quantified Fe and FPN in three separated CA1 layers: stratum molecular–radial (SMR), stratum pyramidal (SP) and stratum oriens (SO). Additionally, chromogenic immunohistochemistry was used to examine the distribution and colocalization with Tau and Aβ proteins. The results show that Fe accumulation was significantly higher in AD brains, particularly in SMR and SO. However, FPN did not present significantly changes in AD, although it showed a non-uniform distribution across CA1 layers, with elevated levels in SP and SO. Interestingly, minimal overlap was observed between Fe and FPN signals, and none between Fe and areas rich in neurofibrillary tangles (NFTs) or neuritic plaques (NP). In conclusion, the lack of correlation between Fe and FPN signals suggests complex regulatory mechanisms in AD Fe metabolism and deposition. These findings highlight the complexity of Fe dysregulation in AD and its potential role in disease progression.

## 1. Introduction

Alzheimer’s disease (AD) is the most common form of dementia, affecting approximately 5–10% of the population over 65 years of age. The pathology involves gliosis as well as synaptic and neuronal loss, leading to the typical symptoms of the disease [1]. Classically, the pathological markers of AD include intraneuronal aggregates of microtubule-associated protein Tau, called neurofibrillary tangles, and insoluble extracellular deposition of β-amyloid peptide (Aβ), which, together with Tau, form neuritic plaques (NP) [1]. In addition, emerging evidence has demonstrated a dysregulation of metal ions (iron, copper and zinc) in the vulnerable brain regions of AD patients [2]. This alteration in the ion content at the tissue level is strongly linked to Aβ deposition, Tau hyperphosphorylation, neuronal loss and neuroinflammation [3]. However, it is not known whether it contributes to the pathogenesis of the disease or is a consequence of it.

Iron (Fe) is a chemical compound necessary for the transport of oxygen in hemoglobin, being part of a significant number of enzymes of the body’s cells. In the central nervous system (CNS), Fe crosses the blood–brain barrier bound to transferrin and accumulates intracellularly thanks to ferritin. In physiological conditions, the excessive amount of intracellular Fe is eliminated by ferroportin (FPN), a transmembrane protein that releases it to the extracellular medium. The quantity of this protein is regulated in turn by hepcidin (HEP) that binds to FPN for cell recycling [4]. Nevertheless, in some pathological situations, the amount of Fe in the brain is altered, which can affect the synthesis of neurotransmitters, the energy balance at the mitochondrial level and the myelination performed by oligodendrocytes [4,5]. The presence of Fe has also been linked to the generation of harmful hydroxyl radicals via Fenton reaction, contributing to increased oxidative stress and neuronal damage [6]. 

In recent years, an increase in Fe in AD brains has been described, mainly in the cerebral cortex, hippocampus and basal ganglia, but also in other areas, such as the cerebellum or brainstem [7]. Notably, Fe accumulation also appears in NP and neurofibrillary tangles in greater quantities than in normal neuropils [8]. Although it is not clear, the increase in Fe amount in the brain of AD patients may be due to the breakdown of the blood–brain barrier or to a deregulation of the proteins related to its accumulation and transport, both intra- and extracellularly. Furthermore, the places where Fe is deposited and the proteins responsible for its transport are still unknown [9]. Since the pioneering work of Bandyopadhyay et al. (2014) introduced the fact that a chelating Fe treatment can prevent the aggregation of misfolded proteins and slow down or reverse the disease process [5], the interest in determining the role of this ion in AD and in the formation of NP and tangles has gained momentum in recent years, even more so when an altered Fe distribution has shown promise as a potential biomarker for early diagnosis in vivo using modern ultra-detection magnetic resonance techniques [10,11]. 

To obtain detailed information about how Fe behaves in situ in samples of human tissues, animal models, or cell cultures, various techniques have been employed to quantify and visualize Fe distribution, ranging from conventional histochemical methods to new bioimaging techniques. Classical histochemical techniques that include staining Fe with Prussian blue or the modifications used by Meguro et al. with DAB [12,13] allow us to obtain spatial images, but they only work for unbound metal ions and do not contribute to a fine quantification. Among new technical approximations, laser ablation inductively coupled plasma mass spectrometry (LS-ICP-MS) stands out for its high sensitivity and cellular resolution [14,15]. LA-ICP-MS uses a focused laser beam to atomize part of a sample. Then, the laser-generated aerosol with the sample extracted using a laser is sent through an ICP, which allows for atomization and ionization of the sample. Subsequently, the ions are separated by their mass–charge ratio in a mass analyzer and finally detected. In this way, isotopic information can be achieved, as well as elemental analysis. Since the laser removes only one area at a time, this technique generates a spatial image of the scanned area with resolution in the low micrometer range. 

Given the observed dysregulation of Fe distribution in AD brains, the aim of this work was to quantify and characterize the microscopic distribution of Fe and FPN within the distinct layers of the hippocampal CA1 region of AD patients via LA-ICP-MS. Additionally, this paper also assessed, using chromogenic immunohistochemistry (IHC), the association between Fe and FPN in relation with Tau pathology and local deposits of Aβ. 

## 2. Materials and Methods

### 2.1. Subjects of Study

The Brain Bank of the Principado of Asturias provided brain tissue samples of patients with AD (*n* = 4) and of healthy humans of various ages (CTRL) (*n* = 4). Table 1 contains the details of the different donors. Patients were clinically diagnosed with AD and the pathology was confirmed post mortem. The different cases were classified based on their AD neuropathological stage, according to Braak’s criteria [16]. The tissue was obtained from necropsies within 6–12 h after death. All grossly visualized and suspected macroscopic or microscopic infarcts were dissected for histologic confirmation by means of Perl’s reaction, and then rejected.

The present study was carried out in accordance with the Helsinki Declaration and approved by the “Regional Clinical Research Ethics Committee of the Principado of Asturias”. These studies received consent on the following bases: (i) samples were collected retrospectively from pathology files of necropsies performed for diagnostic purposes, (ii) patient identities were completely anonymized and unliked from unique identifiers (1–8#) and (iii) there was no risk to the participants (only anonymized tissues were used). 

### 2.2. Tissue Processing

Human hippocampal pieces were fixed by immersion in 10% formaldehyde in 0.1 phosphate buffer (pH 7.4). After fixation, they were washed, dehydrated and embedded in paraffin. Hippocampus cross sections of 5 μm thickness were cut and stretched on slides coated with FLEX (Agilent Technologies, Santa Clara, CA, USA). To avoid formalin-induced artifacts, only material fixed for a period of less than six months was selected. Consecutive sections of each sample were subjected to a corresponding protocol depending on the study, i.e., histochemistry or IHC. Specifically, the CA1 region of the hypothalamus was selected because we had previously reported changes in AD [17], and it is an easily recognizable area in the LA-ICP-MS visor. 

### 2.3. Neurons, Senile Plaques, Neurofibrillary Tangles and Amyloid-Beta Staining

In order to carry out a cytoarchitectonic study of the samples, they were stained using a Nissl-type staining procedure [18]. For the diagnosis and analysis of the histological changes in AD brain tissues, routine procedures such as the silver technique [19] or a Congo red method, developed in our laboratory [20,21], were used to visualize the typical cerebral markers of the neuropathology.

### 2.4. Chromogenic Immunohistochemistry

For conventional chromogenic IHC, human brain sections were deparaffinized, incubated for 60 min at 56 °C and immersed in an antigen retrieval solution (pH 9 at 95 °C for 20 min). Then, tissues were washed three times (10 min) with PBS (10 mM; pH 7.4) and subsequently incubated with 0.1% triton X-100 in PBS (5 min). Next, another sequence of washing steps was performed with PBS (10 min, three times), and the slides were incubated with the blocking agent solution (0.1% BSA and 10% goat serum solution in PBS; 10 mM; pH 7.4) at room temperature for 30 min. Then, consecutive sections were incubated with a rabbit antibody against Aβ (Novocastra NCL-β-amyloid; dilution 1:100) or Tau (Novocastra NCL-Tau-2; dilution 1:500) overnight at 4 °C (Leica Biosystems, Novocastra, St. Gallen, Switzerland). After several washes in PBS, sections were incubated with universal biotinylated horse antibody (Vector, PK-8800; 1:50 dilution; 30 min) (Vector Laboratories Inc., Newark, NJ, USA). Subsequently, sections were incubated with extravidin (Sigma Extra-3, Sigma-Aldrich, St. Louis, MO, USA), and peroxidase activity was visualized by incubation with Sigma Fast DAB (Sigma D4168, Sigma-Aldrich, St. Louis, MO, USA) at room temperature for 30 min. Finally, sections were counterstained using a modified formaldehyde method with thionin [18], dehydrated, clarified in eucalyptol and mounted with Eukitt. Appropriate negative control assays were carried out to ensure that there was a lack of non-specific labeling and amplification. Thus, similar sections were processed in the same way with a non-immune serum or specifically absorbed serum in place of the primary antibody. Under these conditions, no specific immunostaining was observed.

### 2.5. Immunohistochemistry with Gold Nanocluster (AuNC) for LA-ICP-MS

In gold nanocluster (AuNC) IHC, for the study of the amount of FPN with LA-ICP-MS, the FPN antibody was first bound to an AuNC according to a procedure described in a previous work [15]. After blocking, AuNC-labeled anti-FPN antibody (5 μg mL^−^^1^) was added to the sections and incubated overnight at 4 °C. After incubation, a final washing step was performed with PBS, and the slides were stored at −20 °C until LA-ICP-MS analysis. 

### 2.6. Registration of LA-ICP-MS

LA-ICP-MS analyses were performed with a commercial LA system (LSX-213 from Teledyne Cetac Technologies, Omaha, NE, USA) coupled to a dual-focus sector ICP-MS field (Item 2, Thermo Fisher Scientific, Bremen, Germany) working in medium-mass resolution mode to remove polyatomic interferences from the searched isotopes (particularly important for ^40^Ar^16^O interference from ^56^Fe). The commercial ablation cell of the LSX-213 system was replaced with an in-house-built Peltier-cooled ablation chamber with a reduced internal volume that allowed the sample’s temperature to be kept constant at −20 °C. LA-ICP-MS coupling was optimized daily using the SRM NIST 612 glass standard for high sensibility and background intensity, as well as the ^238^U/^232^Th signal ratio, which should be close to 1. The ^248^ThO/^232^Th signal ratio was also measured to control the formation of rust, and was always below 0.5% in the selected optimized conditions. 

Sections of brain tissue were ablated in scanning mode using a laser beam diameter of 10 μm. The overlapping laser points and high repetition rates (20 Hz) resulted in a differential scanning mode, so that better lateral resolution than the selected laser point diameter was achieved (around 5 μm). Samples were ablated line by line under optimized LA-ICP-MS conditions. In all cases, sections of the human hippocampus were scanned with a laser beam using an average of 70 to 80 individual lines. The experimental conditions were optimized in terms of maximum sensitivity and lateral resolution. For image data processing, the contribution of the gas target was removed from the raw intensity signals to work with net intensities. Two-dimensional images obtained by LA-ICP-MS were created using ImageJ 1.49p software (National Institutes of Health, Bethesda, MD, USA). A magnification of the image by a factor of six with respect to the actual size of the image (dimensions µm) was applied and a bicubic interpolation was made to eliminate pixel aspects. 

### 2.7. Preparation of Laboratory Gelatin Standards for Quantitative LA-ICP-MS Measurements

Laboratory gelatin standards were employed to quantitatively analyze the distributions of Fe and FPN (antibody labeled with gold nanoclusters, AuNCs) in sections from human brains. Calibration curves for Au and Fe were created using gelatin standards with concentrations ranging from 0 to 60 μg Au. mL^−^^1^ and from 0 to 52 μg Fe·mL^−^^1^, respectively. These standards were prepared by mixing 15% gelatin (*w*/*w*) with Na·AuCl_4_·H_2_O and FeSO_4_·7H_2_O solutions for Au and Fe calibrations, respectively. The mixture was homogenized by heating and stirring at 60 °C and then frozen at −20 °C. The frozen gelatin standard was cut into 5 µm thick sections and mounted on microscope glass slides, and their Au and Fe concentrations were determined using conventional nebulization ICP-MS after acidic digestion with HNO_3_ and H_2_O_2_ [15].

### 2.8. Statistical Analyses

The data were extracted and transferred to SPSS software version 23 (SPSS, Chicago, IL, USA) for the relevant calculations and statistical analysis. Data were tested for normality of populations and homogeneity of variances. Subsequently, Student’s two-tailed *t*-tests for mean comparison or one- or two-way analysis of variance (ANOVA) tests were employed, followed by Bonferroni’s multiple comparisons test. Differences were considered significant when *p* < 0.05. Data in the graphs are presented as means ± SEM.

## 3. Results

The technique applied here allowed us to visualize and compare two-dimensional images obtained for FPN (^197^Au^+^ signal from Ab-AuNCs) and Fe (^56^Fe^+^) in the CA1 region of the human hippocampus using LA-ICP-MS, under previously reported experimental conditions [11]. This technique provides enough sensitivity to study the distribution pattern of Fe and FPN and to obtain information about its average content in the ablated areas of brain tissue sections of both AD and CTRL. Furthermore, this methodology also enabled us to achieve the mean concentration in selected areas of the two-dimensional images obtained, such as the different layers that form the hippocampus-ablated CA1 region (stratum molecular–radial, SMR; stratum pyramidale, SP; and stratum oriens, SO). 

In Figure 1, qualitative images of Fe distribution in the CA1 region of human hippocampus (AD and CTRL) can be observed, as well as the absolute quantification of the Fe concentration (expressed in µg Fe·g^−^^1^) obtained using LA-ICP-MS in this area. The results demonstrate that human brain tissues exhibited Fe concentrations in the range of 0.3–1.2 µg Fe·g^−^^1^ for CTRL and 2.8–7.1 µg Fe·g^−^^1^ for AD patients. Remarkably, as expected, a statistically significant increase in Fe levels in AD patients (1.09 ± 1.08) was detected compared to CTRL (0.2 ± 0.02 µg g^−^^1^) (Figure 1a). A detailed analysis of the qualitative images showed that Fe ions appeared heterogeneously distributed across the layers of the CA1 region of the hippocampus in both CTRL and AD samples.

The Fe concentration, expressed in µg Fe·g^−^^1^, was calculated for the square-marked area in the qualitative LA-ICP-MS images (see Appendix A). In fact, higher Fe levels were observed in fibers and interneuron areas rather than in pyramidal neurons, which is a novel and important observation (Figure 1b and Figure 2). To assess the value of these differences, we quantified the Fe signal in three areas of the CA1, i.e., SMR, SP and SO. The highest concentrations of Fe generally occurred in the SMR and SO regions of AD brains with statistical signification (Figure 2). It can also be observed that the Fe content was higher in the white matter underlying the layers of gray matter. However, quantification was not feasible, as it exceeded the area ablated by the instrument in the CTRL specimens. 

Regarding FPN, we employed an anti-FPN antibody labeled with gold nanoclusters (Au NCs) to obtain information about the average amount in the SMR, SP and SO layers of the hippocampus CA1 region both in CTRL and AD patients. Protein concentration, expressed in µg Au·g^−^^1^, was calculated for the square-marked area using qualitative LA-ICP-MS images (see Appendix A). In this case, values ranged from 0.6 to 4.5 µg Au·g^−^^1^ in CTRL and from 2.3 to 6.8 µg Au·g^−^^1^ in AD patients. Like the Fe data, there seemed to be a slightly trend of FPN increasing in AD (Figure 3a). Moreover, FPN exhibited non-uniform distribution across the different layers of CA1 (see Figure 3a,b). The highest amount of FPN was observed in SO and SP compared with SMR (Figure 4).

To correlate Fe and FPN localization at the CA1 region, ^56^Fe^+^ and ^197^Au^+^ signals were overlapped in the same image (Figure 5); the Fe distribution can be seen in a green-yellow scale, while the FPN distribution is shown in a purple scale (Figure 5b,f). Images show the qualitative 2D images obtained for the distribution of FPN and Fe in the CA1 region after LA-ICP-MS analysis of an AD (#6) and a CTRL (#4) brain tissue section. Upon observation, it became apparent that there was minimal superposition between the signals of Fe and FPN.

In addition, consecutive sections, comparable to those employed for quantifying Fe and FPN, underwent IHC to assess the presence of the two primary markers associated with AD progression, Tau and Aβ proteins (Figure 5). Our examination revealed minimal staining for Tau and Aβ in the CTRL cases. In contrast, the AD cases exhibited pronounced staining for both proteins, albeit with a more irregular pattern observed for Aβ. This approach aimed to investigate the distribution patterns and establish a comparative analysis of the values of Fe and FPN with respect to Tau and Aβ (Figure 5).

Afterwards, the chromogenic images from IHC were superimposed with those obtained from LA-ICP-MS (Figure 6). In this context, Tau exhibited the highest expression in the SP layer of the hippocampus, forming characteristic NFTs within neurons, both in CTRL brains and, to a greater extent, in AD brains. As shown in the merged image (IHC for Tau and LA-ICP-MS for Fe), no significant amount of Fe was seen in areas with Aβ or a large amount of NFTs (Figure 6e,f). However, a substantial overlap of platelets and NFTs with FPN was found in the pyramidal neuron layer (Figure 6g,h). In the SMR layer with a substantial presence of projections, minimal or no staining for Tau or Aβ was observed, but it was characterized by significant Fe accumulation in both CTRL and AD cases (Figure 6a–f). As mentioned above, gray–white matter boundaries were aligned with elevated Fe levels, but reduced FPN levels.

It is noteworthy that, occasionally, large Fe-filled structures corresponding to large blood vessels could be observed in the bioimage when compared with the instrument camera image; it is advisable to take this into consideration (see Appendix A for details).

## 4. Discussion

The present study takes advantage of the LA-ICP-MS technique to visualize and compare two-dimensional images of FPN and Fe locations in the CA1 region of the human hippocampus in AD patients. Notably, this technique proved to be sensitive enough to study the distribution patterns of Fe and FPN, and also to provide information about its concentrations in specific CA1 layers such as SMR, SP and SO, enhancing the understanding of the regional distribution of Fe and its cellular transporter, FPN. The LA-ICP-MS alone allowed us to visualize and quantify tissue metal ions, but with the methodological improvement applied here, we could also obtain images of specific proteins in biological tissues using metal-labeled antibodies (gold in our case). This could even be combined with an ICH protocol, which constitutes (with certain limitations of spatial resolution) an interesting alternative to the detection of fluorescence or chromogenic signals, which are commonly used [14,15].

Classical immunofluorescence histochemistry, employed as a method of detection of molecules in the nervous system, has an important limitation; part of the fluorescent emission observed in the images can be attributed to the presence of lipofuscin in the brain. Lipofuscin is an autofluorescent pigment made up oxidized lipids and proteins that accumulates in postmitotic cells with age. It is especially abundant in neurons, and much more so in humans than in other vertebrates. There are treatments to mask the fluorescent emission of lipofuscin (e.g., Sudan Black B); however, such techniques also block the fluorescence of the IHC signal and can only be used when the antigen is expressed in high concentrations [22,23]. This is not the case with the FPN, so this new bioimaging technique has an additional advantage. As demonstrated in this work, metallic NCs could be used as metal ions for analysis and as fluorescent labels for mapping biomolecules of interest in biological tissues, replacing conventional fluorophores. In our case, NCs allow us to achieve double bioimaging of metals by avoiding the lipofuscin fluorescence [24]. Moreover, the use of different NCs to label different analytes is essential for multiparametric analysis [15,25]. 

Excessive accumulation of Fe in specific brain regions is increasingly being related to AD [7]. In recent decades, many techniques have been used to assess the amount of Fe in the AD brain, and each of these can give rise to biases [26,27]. For example, post mortem Fe testing has the advantage of providing direct measurements of this ion in the brain, which is currently not possible in living people despite the improvements in magnetic resonance imaging (MRI) measurements [26,28]. Quantitative studies based on histological staining (Perl’s and Turnbull stain) are imprecise, but offer the possibility of visually evaluating sections at cellular resolution. Specifically, Fe staining allows users to selectively dye non-heme Fe^3+^, Fe^2+^ or both [12,13]. By contrast, most physical techniques quantify all the different forms of Fe without distinction [29]. Despite some limitations, such as a reduced spatial resolution that may hinder detailed examination of microstructural features, or the lack of molecular information about the chemical forms of Fe present in the tissue, our post mortem approach offers valuable quantitative insights into nervous tissue, providing marker distribution maps for bioimaging which can be complemented with other techniques to ensure a better understanding of the role of Fe in AD. Notably, with the technique used here, we lost precision iron ionic, but gained quantitative and cytoarchitectonic ones.

The presence of Fe both in lower and upper nervous centers during aging and AD is undeniable, albeit at varying levels [7]. There is also evidence demonstrating the association between Fe accumulation in the brain and cognitive deficits in AD [30]. Previous studies have linked Fe enrichment to the progression of AD and the increasing presence of Tau and Aβ deposits in the cortex [31,32]. In our samples, no significant changes in Fe concentration were observed in the pyramidal stratum despite the presence of NP and numerous NFTs in the CA1 layers of AD brains. The evidence regarding the association between Fe and AD’s pathology remains inconclusive for amyloid plaques [12,31,33,34], with only a limited association found with tangles [35,36]. It must be added that, in several reports, the images were undetermined, possibly due to the relatively low Fe content in these specific areas [36,37]. Currently, it is challenging to determine whether elevated levels of Fe and other metals are related to the formation and progression of these aggregates or whether it is a consequence of the Fe increase within the brain parenchyma. It is important to consider that Fe deposition in these pathological features may represent only a small part of the total accumulated Fe load.

Although most studies have focused on the relationship between Fe and pathology-associated proteins in gray matter, both in vivo and post mortem images have demonstrated that the amount of deposited Fe is higher in white matter than in gray matter in both control and AD subjects, at least in the neocortex [11,12,38]. We found that the gray–white matter boundary in SMR coincides with elevated Fe levels in the hippocampus of AD patients. But we have not observed a relationship between layers with a high Fe content and those where the formation of NFT and amyloid plaques is concentrated. Therefore, the role of this ion in the formation of these characteristic features of Alzheimer’s disease remains difficult to determine. Perhaps these small amounts of intra- or extracellular Fe may be capable of causing significant cellular damage due to reduced capacity in these areas to combat free radicals.

In the AD research field, it remains unclear whether increases in Fe are due to a failure of its usage (excessive Fe storage), an alteration of the clearance processes (transporter defect) or both. The study of expression of proteins that maintain systemic Fe balance could help in this way. Transferrin (Tf) is the mayor Fe transporter across the blood–brain barrier by Tf receptors (TfR1) that mediates Tf-Fe endocytosis of the endothelium [39]. Another membrane Fe transporter found at the systemic level is divalent metal transporter 1 (DMT1). The presence of TfR1 and DMT1 on the membranes of neurons, astrocytes, oligodendrocytes and microglia suggests that all these brain cells have the capacity to absorb Fe [39]. Furthermore, accumulated evidence demonstrates that Fe efflux from them occurs through a pathway involving FPN. It has been described that the dysregulation of Fe homeostasis in neurodegenerative diseases may depend on the absence of or reduction in expression of Fe exporters, which could increase the amount of accumulated Fe in the brain [39,40]. We investigated FPN expression and found a slight increasing trend in AD cases, but this increase was not statistically significant. This could be due to the small number of cases, so a more extensive study would be necessary, with the difficulty of finding homogenized processing samples that this entails. However, we observed that FNP is not distributed homogeneously in the CA1 layers, with higher values in the SO and SP layers and lower values in SMR, which corresponds to the areas in which there are a greater number of nerve cells. Some reports have investigated FPN using classical IHC in the human hippocampus, entorhinal cortex and superior frontal gyrus and found reduced labeling assessed using densitometric analysis, which is a less accurate method [4,41]. The same results were found in steady-state protein levels measured by Western blotting using brain tissue (cortex including hippocampus and dentate gyrus) from an animal model of AD (APP-tg mice) [4,41]. The authors showed that FPN is present in normal neurons and astrocytes and could be reduced with age and AD, but more studies are needed to obtain more reliable results [41,42].

It has been suggested that deregulation of FPN could lead to an abnormal accumulation of intracellular Fe, exacerbating oxidative stress and accelerating the neurotoxicity and neurodegeneration associated with AD, but with our results, we cannot corroborate this hypothesis. To gain insight into the correlation between Fe and FPN, signals were overlapped in the same image within the CA1 region. The analysis revealed minimal superposition between them, suggesting an independent distribution. In fact, our data demonstrate that the quantity of FPN in pyramidal cells and interneuron layers does not significantly vary with the disease, suggesting that this protein may still be playing a role in maintaining Fe release, attempting to keep both intra- and extracellular levels low in the region. In contrast, Fe levels increase in AD, particularly in SMR and SO, where there is quantitatively less FPN. This is somehow associated with elevated Fe levels in the tissue parenchyma surrounding projection zones and both myelinated and non-myelinated axons. The excess Fe could be compromising communication and myelination in the area, as is suggested to occur in the white matter.

Beyond the involvement of Fe transporters, recent studies suggest that a breakdown of the blood–brain barrier could be the cause of the progressive accumulation of this ion in the brain parenchyma during aging and AD [43]. On the one hand, dysfunction in blood–brain barrier transport systems lead to the development of Aβ and Tau pathology and neuronal loss. Taking into consideration that the transcellular transport of molecules across endothelial cells may be compromised, it is conceivable that Tf and other Fe-transporting molecules could infiltrate and persist within the neuropil, ready for Fe cellular uptake. On the other hand, this vascular process may be responsible for the progressive reduction in cerebral blood flow, triggering oxidative stress and the subsequent death of astrocytes and neurons, particularly in energy-demanding brain regions like the cortex and hippocampus [43,44]. These changes may be the cause of the increase in Fe that precedes the accumulation and aggregation of Aβ plaques and hyperphosphorylated Tau tangles.

This work also offers a remarkable new perspective on the localization of Fe within the various layers of CA1. While previous attention has been largely directed towards the pyramidal neuron layer, our study did not reveal particularly striking changes in this region. Conversely, the layers containing interneurons and more extensive projections accumulate higher Fe levels in patients with AD. We attempted to elucidate this phenomenon by examining the expression of the cellular Fe exporter FPN, but the data obtained did not provide clear conclusions. It is not possible to determine with this technique whether it is the glial cells or the neurons that maintain FPN levels in the disease. However, what seems clear is that the accumulation of Fe in the hippocampus in AD must have another source or cause besides the dysregulation of the FPN pathway. If it is demonstrated that this ion accumulates pathologically in nervous tissue independently of the failure of the Fe transport/storage system, Fe chelator-based treatments would constitute a promising therapy to combat AD and other related diseases.

These comprehensive results provide valuable insights into the intricate relationships among Fe, FPN and classical markers of AD pathology within the CA1 region of the hippocampus. The non-homogeneous distribution patterns and the minimal correlation between Fe and FPN signals suggest the presence of underlying mechanisms and relatively unexplored regulatory processes governing Fe accumulation and its involvement in the progression of AD. Further research, including the ablation of larger tissue samples, is needed to validate and expand the current findings as well as to fully elucidate the cause and consequences of Fe accumulation as a potential target for therapeutic interventions. 

## 5. Future Perspectives

The LA-ICP-MS technique, combined with metal-labeled antibodies, described herein opens new possibilities for improving bioimaging techniques to visualize and quantify not only ions, but also specific proteins in biological tissues. Future research should focus on refining this methodological strategy to obtain more detailed images at the cellular level that provide a better understanding of the distribution and relationship between ion compounds and certain proteins, such as Tau and Aβ, in the context of AD brains. Finally, to obtain a more complete and accurate image of the distribution of Fe and/or other metals in the brain, it would be interesting to validate the results obtained in this study using other diagnostic techniques such as MRI, which would complement the measurements obtained with LA-ICP-MS.

## Figures and Tables

**Figure 1 biomolecules-14-00295-f001:**
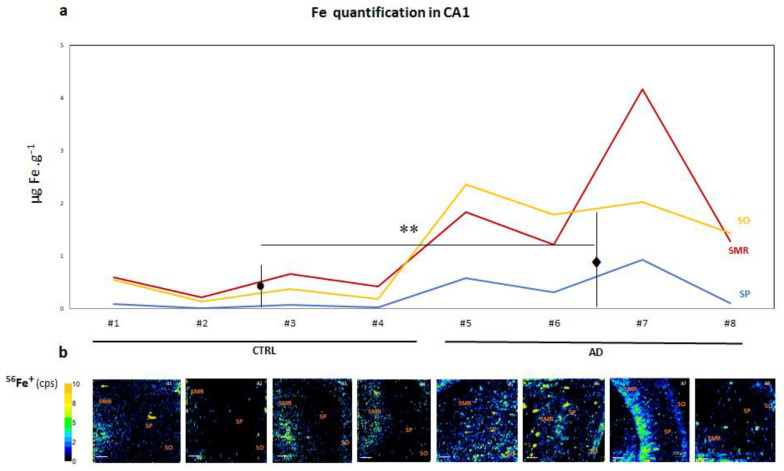
(**a**) Line graph corresponds to average ^56^Fe^+^ concentration obtained in three regions of CA1, stratum molecular–radial (SMR) in red, stratum pyramidale (SP) in blue and stratum oriens (SO) in yellow, from the human hippocampus CA1 region in CTRL (donors #1, #2, #3 and #4) and AD (donors #5, #6, #7 and #8) using LA-ICP-MS (expressed as µg Fe per g^−1^ of tissue). The average mean values ± SEM for the CTRL and AD cases are depicted with black dots and diamonds in the image, respectively. Significant differences were analyzed using Student’s *t*-tests. ** *p* < 0.001 compared to control. (**b**) The qualitative images obtained for the Fe signal in each CA1 hippocampal sample can be seen below the graph. The color scale refers to the amount of Fe in the image, with the greatest amount in orange-yellow and the lowest amount in black-dark blue. Cps: counts per second. Bars represent 200 µm.

**Figure 2 biomolecules-14-00295-f002:**
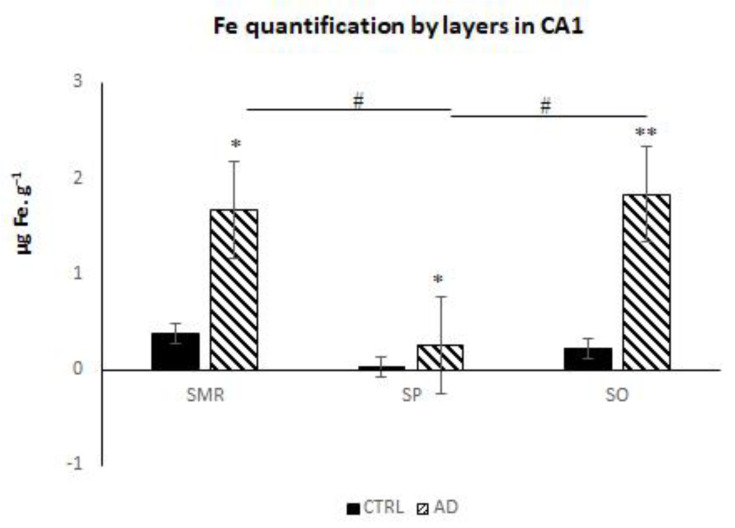
Fe quantification (Fe micrograms per gram) via LA-ICP-MS in the different layers of human hippocampus CA1 in AD and CTRL samples: stratum molecular–radial (SMR), stratum pyramidale (SP) and stratum oriens (SO). Bars represent average mean ± SEM. Significant differences were analyzed using two-way ANOVA followed by Bonferroni’s post hoc test. ^#^ *p* < 0.05, ** *p* < 0.001, * *p* < 0.05.

**Figure 3 biomolecules-14-00295-f003:**
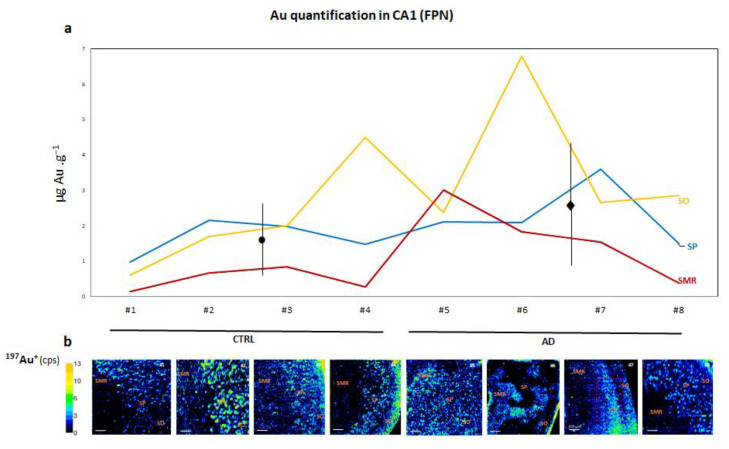
(**a**) Line graph corresponds to average ^197^Au^+^ concentration (FPN) obtained in three regions of CA1: stratum molecular–radial (SMR), stratum pyramidale (SP) and stratum oriens (SO) of human hippocampus CA1 in CTRL (donors #1, #2, #3 and #4) and AD (donors #5, #6, #7 and #8) using LA-ICP-MS (expressed as µg FPN per g^−1^ of tissue). The average mean values ± SEM for the CTRL and AD cases are depicted with black dots and diamonds in the image, respectively. (**b**) Qualitative images obtained by LA-ICP-MS for ^197^Au^+^ distribution using AuNCs bioconjugated with FPN antibody at 5 µg mL^−1^ concentration can be seen below the graph. The color scale refers to the amount of Au in the image, with the greatest amount in orange-yellow and the lowest amount in black-dark blue. Cps: counts per second. Bars represent 200 µm.

**Figure 4 biomolecules-14-00295-f004:**
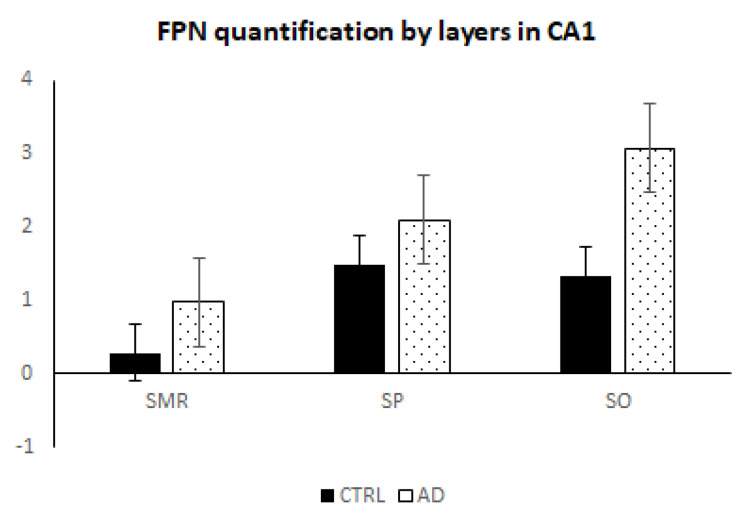
FPN quantification (Au micrograms per gram) by LA-ICP-MS in the different layers of human hippocampus CA1 of AD and CTRL samples: stratum molecular–radial (SMR), stratum pyramidale (SP) and stratum oriens (SO). Bars represent average mean ± SEM. Significant differences were analyzed using two-way ANOVA followed by Bonferroni’s post hoc test.

**Figure 5 biomolecules-14-00295-f005:**
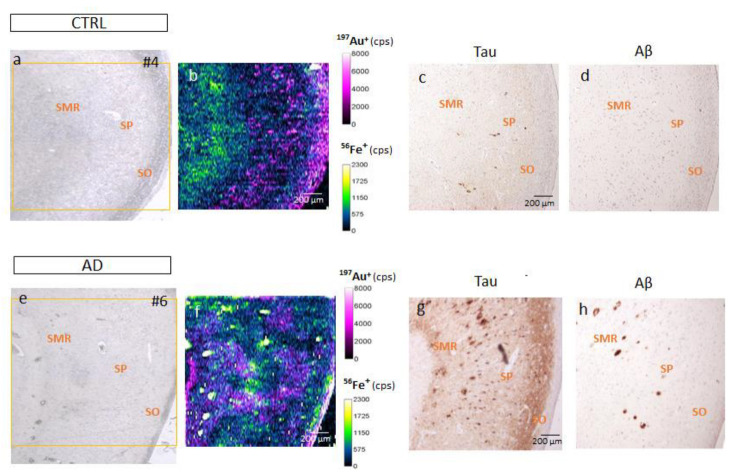
Comparison of LA-ICP-MS and IHC images of amyloid-β (Aβ) and Tau in consecutive sections in selected samples of CTRL (#4) and AD (#6). (**a**,**e**) Transmission image obtained for selected region (yellow box) in the section of hippocampus CA1 with camera of the laser system after an IHC with FPN-AuNCs. (**b**,**f**) Qualitative image obtained using LA-ICP-MS for ^56^Fe^+^ and ^197^Au^+^ (FPN). Distributions of both elements were overlapped for comparison. (**c**,**g**) IHC for Tau revealed with DAB chromogen in consecutive section. (**d**–**h**) IHC for Aβ revealed with DAB chromogen in consecutive section. Bars represent 200 µm.

**Figure 6 biomolecules-14-00295-f006:**
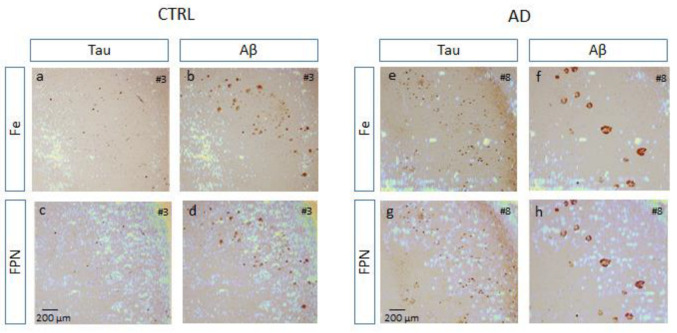
Superposition of images between those obtained by classic IHC for Tau and β-amyloid (Aβ) and those by LA-ICP-MS in one control (CTRL) (#3) and other Alzheimer’s disease (AD) sample (#8). (**a**) Tau + Fe; (**b**) Aβ + Fe; (**c**) Tau + FPN; (**d**) Aβ + FPN of CTRL sample (#3). (**e**) Tau + Fe; (**f**) Aβ + Fe; (**g**) Tau + FPN; (**h**) Aβ + FPN of AD sample (#8). Cortical gray matter from hippocampal area CA1 shows low staining for Aβ (**b**,**d**), Tau, and iron in the control sample, but high FPN in the SP and SO layers. Gray-white matter boundaries coincide with high iron staining (SMR) but low FPN. In the AD sample, the SO layer in the gray matter has moderate iron staining but low Aβ and Tau staining. The SMR slayer has strong Aβ and Tau staining corresponding to images with high iron content. The SP layer has the lowest iron levels but the highest FPN concentrations. Bars represent 200 µm.

**Table 1 biomolecules-14-00295-t001:** Details of AD and CTRL human brain tissues from postmortem donors (#).

Donor	Age	Sex	Cause of Death	Break Scale	Other Brain Pathology
#1	57	Male	Cardiogenic shock	--	No
#2	71	Female	Cardiorespiratory arrest	--	No
#3	82	Male	Cardiorespiratory arrest	--	No
#4	100	Female	Multi-organic failure	--	No
#5	84	Female	Multi-organic failure	II–III	No
#6	77	Male	Cardiorespiratory arrest	VI	No
#7	91	Female	Acute myocardial infarction	V–VI	No
#8	62	Male	Cardiorespiratory arrest	VI	No

## Data Availability

The datasets generated during and/or analysed during the current study are not publicly available due to privacy/ethical restrictions but are available from the corresponding author on reasonable request.

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
