# Peer review of "Iron Dysregulation in Alzheimer’s Disease: LA-ICP-MS Bioimaging of the Distribution of Iron and Ferroportin in the CA1 Region of the Human Hippocampus"

_biomolecules, 2024, doi:10.3390/biom14030295_

Round 1

Reviewer 1 Report

Comments and Suggestions for Authors

This promising study approaches a subject of major relevance with scientific soundness.

The authors presented well-organized sections, clearly stated objectives, and rationally discussed findings. I have no hesitation in recommending it for acceptance. During my reading, I identified a few points that could be better addressed to enhance clarity, which follows:

1) If possible, TAU and beta AM contents should be expressed in terms of positive cells;

2) A paragraph on the study's main limitations would nicely complement the manuscript for proper data interpretation. The authors are also invited to discuss the potential impact of the sample size on the data interpretation;

3) Future perspectives should be included;

4) The authors could mention how they picture that these findings may contribute to managing AD;

5) Page 11, line 419 - this paragraph seems incomplete. Please verify it;

Author Response

This promising study approaches a subject of major relevance with scientific soundness. The authors presented well-organized sections, clearly stated objectives, and rationally discussed findings. I have no hesitation in recommending it for acceptance.

Response: we appreciate the positive comments.

During my reading, I identified a few points that could be better addressed to enhance clarity, which follows:

1) If possible, TAU and beta AM contents should be expressed in terms of positive cells.

Response: we appreciate the comment. It is true that we could have explained our results based on the number of Tau and/or Aβ positive cells. However, it has been clearly demonstrated that immunolabeling of these proteins is not limited to neuronal bodies, but appears in neuronal extensions and therefore in the entire neuropil. Honestly, we think that our way of expressing the results provides more information and fits better with the final aim of the work, which is to decipher the connection between Fe and FPN in relation to Tau pathology and local Aβ deposits.

2) A paragraph on the study's main limitations would nicely complement the manuscript for proper data interpretation. The authors are also invited to discuss the potential impact of the sample size on the data interpretation.

Response: thanks for raisin this issue that has been considered in the revised version of the manuscript.

3) Future perspectives should be included.

Response: we appreciate this suggestion. As per reviewer’s request a future perspectives section has been included in the revised version of the manuscript.

4) The authors could mention how they picture that these findings may contribute to managing AD.

Response: we acknowledge this comment that has been taken into account in the revised version of the manuscript. In this sense, a new paragraph about how our findings contributing to the knowledge of AD, emphasizing the practical implications in current therapies, has been included.

5) Page 11, line 419 - this paragraph seems incomplete. Please verify it.

Response: thanks you very much for detecting this mistake (fixed in the revised version of the manuscript).

Observation: Major changes are highlighted in yellow. Grammatical corrections are not highlighted.

Reviewer 2 Report

Comments and Suggestions for Authors

In this paper, for the first time, the distribution of Fe and FPN in the CA1 region of AD brains was analyzed using the latest analytical method, LA-ICP-MS imaging. The results revealed that Fe levels in the CA1 region of the AD brain were significantly higher than those in normal controls, while FPN levels were not significantly different in that brain region. The results also showed that the increased Fe signals did not correlate with tau lesions, indicating that there is a complex Fe metabolic regulatory mechanism in the brain, and that metabolic abnormalities may be involved in the progression of AD. The analysis of the pathophysiology of AD using these state-of-the-art technologies is novel and will contribute to the elucidation of the true nature of AD and the development of treatment methods. Therefore, this study deserves to be published in Biomolecules.

Author Response

In this paper, for the first time, the distribution of Fe and FPN in the CA1 region of AD brains was analyzed using the latest analytical method, LA-ICP-MS imaging. The results revealed that Fe levels in the CA1 region of the AD brain were significantly higher than those in normal controls, while FPN levels were not significantly different in that brain region. The results also showed that the increased Fe signals did not correlate with tau lesions, indicating that there is a complex Fe metabolic regulatory mechanism in the brain, and that metabolic abnormalities may be involved in the progression of AD. The analysis of the pathophysiology of AD using these state-of-the-art technologies is novel and will contribute to the elucidation of the true nature of AD and the development of treatment methods. Therefore, this study deserves to be published in Biomolecules.

Response: we appreciate the clear and succinct summary of our work, and the positive evaluation of the manuscript.
